# Boosting Image Dehazing via Elaborate Integration of Complementary Dependencies

## Abstract

Haze removal seeks to restore clear images from hazy inputs. Previous research demonstrates that short-range dependencies are effective for preserving local details, while long-range dependencies capture global context. Because both are essential to dehazing and complement each other, many approaches explicitly integrate them within dual-stream frameworks. However, *the trustworthy aggregation of these dependencies remains underexplored*. In this paper, to optimize the contributions of dependencies at varying ranges, we first conduct comprehensive quantitative and qualitative experiments to identify the key influencing factors. Our findings indicate that an effective aggregation strategy should jointly consider haze density and semantic information. Building on these insights, we introduce a CLIP-enhanced Dual-Path Aggregator for the class of dual-stream dehazing methods. This module first employs a shared backbone to generate fine-grained haze density and semantic maps in a computationally efficient manner, and then uses them to instruct the integration process. Extensive experiments show that the proposed aggregator significantly improves the performance of existing dual-stream methods, and our custom-built model, DehazeMatic, achieves state-of-the-art results across multiple benchmarks. As an additional contribution, we also address, for the first time, the challenge of accurately estimating haze density maps.

## 1 Introduction

Image dehazing serves as an essential pre-processing step for high-level vision tasks in hazy environments, such as object detection Li et al. (2023) and semantic segmentation Ren et al. (2024).

Recent data-driven approaches can be broadly divided according to the receptive field size of their feature extractors: (*i*) using convolution Bai et al. (2022); Cai et al. (2016); Dong et al. (2020); Li et al. (2017); Ren et al. (2018; 2020); Zhang & Patel (2018) or window-based self-attention Kulkarni et al. (2022); Kulkarni & Murala (2023); Song et al. (2023); Wang et al. (2023), which capture fine-grained local structures but struggle with holistic reasoning Kim et al. (2023); Veit et al. (2016); De & Smith (2020); and (*ii*) using linear self-attention Qiu et al. (2023) or state-space models (SSMs) Shen et al. (2023); Zhou et al. (2024); Zhang et al. (2024a), which excel at modeling long-range dependencies yet often sacrifice 2D inductive biases Huang et al. (2024).

Motivated by the fact that both types of methods have shown strong performance and their respective strengths can offset each other's weaknesses, recent work has explored dual-stream architectures that explicitly integrate short- and long-range cues Zamir et al. (2022); Chen et al. (2024a); Jiang et al. (2024); Liu et al. (2024a). However, rationally aggregating information from these two streams remains a non-trivial task, as tokens with different characteristics in an image vary in their need for local detail versus global semantics (*i.e.*, the trade-off between short- and long-range dependencies). Existing methods typically rely on simple operations, such as addition, concatenation, or self-learned gating network, which hinder the optimal utilization of both dependency types, thereby creating performance bottlenecks. Ultimately, this stems from the lack of clear guidance on how to assign appropriate importance to short- and long-range dependencies on a per-token basis.

To fill this gap, we begin with a quantitative and qualitative analysis of the key factors governing this trade-off and ultimately find that haze density and semantic information play decisive roles. The subsequent objective is to accurately estimate a haze density map and a semantic information map

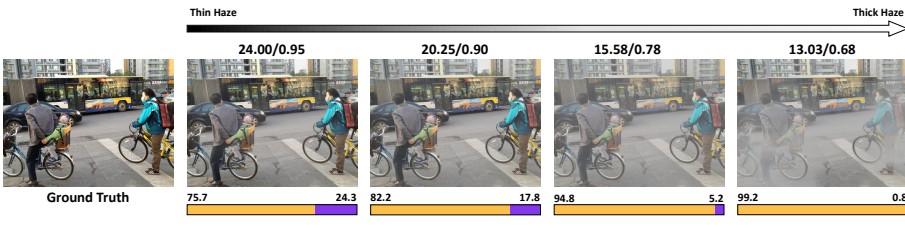

Figure 1: Illustration of CLIP Radford et al. (2021)'s potential to perceive haze and its density. We use the ViT-B/32 OpenCLIP Ilharco et al. (2021) model pre-trained on the LAION dataset. Values above images are PSNR/SSIM (quantifying haze density), and values below are CLIP similarity scores with paired prompts. As density increases, similarity with haze-describing prompt also rises.

to instruct the integration process. Moreover, to enhance model efficiency, we further aim to derive both maps from a shared backbone.

However, this objective remains challenging: not only is there no method capable of accurately estimating haze density map, but extracting two types of information with substantial modality differences from a single backbone remains inherently difficult. Inspired by recent advances in CLIP Radford et al. (2021), pretrained on web-scale datasets and capable of encoding rich semantic priors that support strong zero-shot semantic segmentation performance Zhou et al. (2023); Zhang et al. (2024c), we further observe that CLIP has the potential to perceive haze and its density, as illustrated in Figure 1. Building on this insight, we propose the CLIP-enhanced Dual-path Aggregator (CedA), a plug-and-play module designed to replace the naïve aggregators used in existing dual-stream dehazing methods. By freezing the CLIP image encoder and training only a set of learnable prompt tokens, CedA simultaneously extracts accurate patch-wise haze density maps and semantic maps from image embeddings. These two maps only need to be passed through a lightweight linear layer to generate aggregation weights, thereby enabling the model to efficiently and adaptively integrate short- and long-range dependencies and achieve substantial performance gains. Finally, to demonstrate that the proposed CedA module can enhance dual-stream dehazing networks and achieve promising results, we develop a benchmark model, DehazeMatic, and conduct extensive experiments. Our contributions are threefold:

- We are the first to identify the key factors governing the relative importance of short- and long-range dependencies in image dehazing. Building on this insight, we present a plug-and-play, trustworthy, and general aggregation module for existing dual-stream dehazing methods, enabling more effective utilization of both short- and long-range cues.

- We introduce DehazeMatic, a benchmark dual-stream dehazing model that achieves *state-of-the-art* performance across multiple datasets and showcases the untapped potential of dual-stream designs for haze removal.

- We further explore the potential of CLIP in haze removal and, for the first time, achieve accurate estimation of haze density maps, without any fine-tuning of the encoder.

## 2 RELATED WORK

### 2.1 SINGLE IMAGE DEHAZING

Image dehazing is an ill-posed problem due to spatially variant transmission and atmospheric light. Early prior-based methods He et al. (2016); Fattal (2008); Kim et al. (2019); Tan (2008); Zhu et al. (2015); Berman et al. (2018) relied on assumptions to estimate parameters in the Atmospheric Scattering Model Narasimhan & Nayar (2002), but often failed when images deviated from these priors.

With the rapid advancement of deep learning Krizhevsky et al. (2012), various learning-based methods have been proposed, resulting in improved performance. Early methods Cai et al. (2016); Li et al. (2017) employ neural networks to estimate key parameters in the ASM and subsequently restore the haze-free images. Later, ASM-independent deep networks Ren et al. (2016; 2018); Liu et al. (2019); Li et al. (2019); Shao et al. (2020); Dong et al. (2020); Zhang et al. (2020); Qin et al.

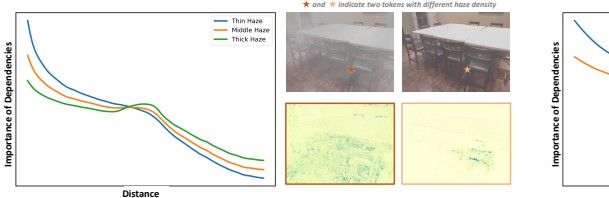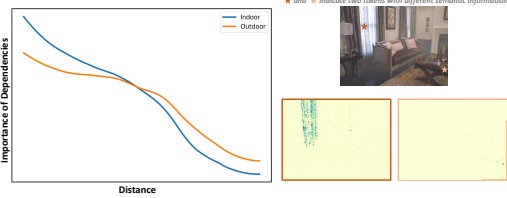

Figure 2: **Illustration of factors influencing the relative importance between short-range and long-range dependencies**. The left panel shows how quantitative results vary with different characteristics of the hypothesized factors. The horizontal axis denotes the Euclidean distance between a token and all others (*i.e.*, the dependency range), while the vertical axis indicates the average importance of dependencies at that distance, averaged over all tokens in the test set. The right panel visualizes the importance of other tokens for selected ones (marked with pentagrams). In (a), the tokens occupy the same location in the same ground-truth image but differ in haze density. In (b), they share a uniform haze level but differ in semantics within the same image.

(2020); Li et al. (2020); Wu et al. (2021); Ye et al. (2022); Song et al. (2023); Feng et al. (2024); Yang et al. (2024b); Zhang et al. (2024b); Chen et al. (2024b); Fang et al. (2024b); Cong et al. (2024); Wang et al. (2024b); Yang et al. (2025); Cui et al. (2025) directly estimate clear images or haze residuals. DehazeFormer Song et al. (2023) is a representative method that achieves efficient feature extraction through window-based self-attention and several key modifications. However, its inherently limited receptive field constrains its performance potential. To obtain a global receptive field with low computational cost, several approaches introduce linear self-attention Qiu et al. (2023), frequency-domain information Shen et al. (2023); Yu et al. (2022), or the Mamba Zheng & Wu (2024); Zhang et al. (2024a) architecture into image dehazing. Many methods Zamir et al. (2022); Chen et al. (2024a); Jiang et al. (2024); Zhang et al. (2025); Liu et al. (2024a) design dual-stream networks that explicitly integrate short- and long-range dependencies, harnessing their complementary strengths to achieve high performance. However, they often overlook the need for effective aggregation across different ranges, resulting in suboptimal outcomes.

## 2.2 CLIP FOR LOW-LEVEL VISION TASKS

Classic vision-language models like CLIP Radford et al. (2021), aim to learn aligned features in the embedding space from image-text pairs using contrastive learning. Some studies have explored leveraging the rich prior knowledge encapsulated in CLIP to assist with low-level vision tasks.

In All-in-One image restoration, some researchers Luo et al. (2023); Ai et al. (2024); Jiang et al. (2025) use degradation embeddings from the CLIP image encoder to implicitly guide networks in making adaptive responses. For monocular depth estimation, recent studies Zhang et al. (2022); Auty & Mikolajczyk (2023); Hu et al. (2024) employ CLIP to map input patches to specific semantic distance tokens, which are then projected onto a quantified depth bin for estimation. In low-light enhancement, some methods Liang et al. (2023); Morawski et al. (2024) use text-image similarity between the enhanced results and learnable prompt pairs to train the enhancement network.

Some works also integrate CLIP into image Wang et al. (2024a) and video Ren et al. (2024) dehazing, but mainly use text-image similarity between dehazed results and contrastive prompt sets as a regularizer. In contrast, our method further exploits the potential of CLIP by directly incorporating its latent embeddings into the main network to guide the dehazing process.

## 3 MOTIVATIONAL EXPERIMENT

Optimal aggregation of dependencies is essential for a dual-stream network to fully leverage short- and long-range cues for dehazing. To this end, we empirically investigate the key factors that govern their relative importance, thereby enabling more reasonable and effective aggregation.

## 3.1 QUANTIFYING THE IMPORTANCE OF DEPENDENCIES

Dependency denotes the influence exerted by other tokens on the current token Bengio et al. (1994); Hochreiter & Schmidhuber (1997). For the experiment presented in this section, we train a Transformer model with a global receptive field, and define the importance of a dependency as the attention weight that another token assigns to the current token in the self-attention mechanism, while its range is measured by the Euclidean distance between the two tokens.

## 3.2 EXPERIMENTAL DESIGN

Our experiment adopts a hypothesis-driven approach, in which we first identify potential key factors and then verify their validity through both quantitative and qualitative analyses. Specifically, we sample image tokens exhibiting diverse characteristics with respect to the hypothesized factors and examine whether the importance of dependencies at a fixed distance *varies* accordingly; the results are presented in Figure 2. The quantitative measure used for each point on the curve is defined as:

$$I(r;c) = \frac{1}{|S(I)|} \sum_{(u,v) \in S(I)} \left( \frac{1}{|B_{(u,v)}(r)|} \sum_{(p,q):\, d_{(u,v),(p,q)} \in r} \widetilde{A}_{(u,v),(p,q)} \right)$$

$$\text{where} \quad \widetilde{A}_{(u,v),(p,q)} = \frac{A_{(u,v),(p,q)}}{\sum\limits_{(\hat{p},\hat{q})} A_{(u,v),(\hat{p},\hat{q})}}. \tag{1}$$

Here, $I(r;c)$ denotes the mean importance of dependencies at distance $r$ under condition $c$, where $c$ represents the dataset characteristic associated with each curve (*e.g.*, haze level or semantic category). $A_{(u,v),(p,q)}$ is the $L_1$-normalized attention weight of the token at $(p,q)$ with respect to the token at $(u,v)$. $d_{(u,v),(p,q)}$ denotes the Euclidean distance between these tokens, and $B_{(u,v)}(r)$ is the set of tokens $(p,q)$ whose distance from $(u,v)$ satisfies $d_{(u,v),(p,q)} \in r$. Finally, $S(I) = \{(u,v) \mid u = 1, \ldots, H;\ v = 1, \ldots, W\}$ is the set of all token coordinates in an image, where $H$ and $W$ denote the image height and width, respectively. The qualitative results in Figure 2 visualize $\widetilde{A}_{(u,v),(p,q)}$ over all possible locations $(p,q)$ with respect to the anchor token $(u,v)$.

## 3.3 EXPERIMENTAL OBSERVATIONS

**Haze density** is intuitively regarded as a key contributing factor. To validate this, we synthesize hazy images with varying density levels using the Atmospheric Scattering Model Narasimhan & Nayar (2002) and conduct corresponding experiments. As illustrated in Figure 2(a), as the haze becomes denser, the relative importance of long-range dependencies increases, while that of short-range dependencies decreases—and vice versa. Quantitative results further confirm this observation.

**Semantic information** is also hypothesized to be an influential factor, as prior work Huang et al. (2020) indicates that scenes with different levels of complexity require dependencies at varying ranges. To examine this, we conduct experiments on indoor and outdoor images from the RESIDE dataset Li et al. (2018), which generally exhibit distinct semantic characteristics. As shown in Figure 2(b), the quantitative results support our hypothesis, while qualitative results further demonstrate that for tokens with different semantic content within the same image, the relative importance of dependencies at different ranges also differs.

Based on these findings, we are the first to propose that, in dehazing, the relative importance of short- and long-range dependencies is jointly influenced by both *haze density* and *semantic information*.

## 4 METHOD

### 4.1 OVERVIEW OF CLIP-ENHANCED DUAL-PATH AGGREGATOR

The proposed CedA is designed to replace the naïve aggregators in existing dual-stream dehazing networks and thereby elevate their performance. Inspired by the observations in Section 3, CedA

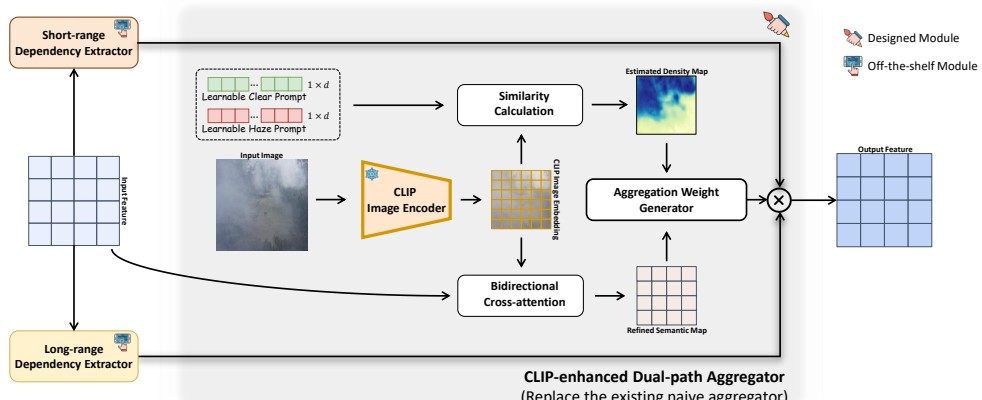

Figure 3: We present a plug-and-play, trustworthy dual-path aggregator, termed CLIP-enhanced Dual-path Aggregator (CedA), designed for dual-stream image dehazing networks.

first generates fine-grained haze density and semantic maps, and then produces pixel-level weights accordingly to adaptively aggregate the outputs from the two paths. Its formulation is given by:

$$\mathbf{F_{out}} = \mathcal{W}\big(\mathcal{H}, \mathcal{S}\big) \odot \mathscr{D}_{long}(\mathbf{F_{in}}) + \Big(\mathbf{1} - \mathcal{W}\big(\mathcal{H}, \mathcal{S}\big)\Big) \odot \mathscr{D}_{short}(\mathbf{F_{in}}) \tag{2}$$

Here, $\mathbf{F_{in}}, \mathbf{F_{out}} \in \mathbb{R}^{H \times W \times C}$ denote the input and output of the core building block, respectively, which consists of a long-range dependency extractor $\mathscr{D}_{long}$ and a short-range dependency extractor $\mathscr{D}_{short}$. $\mathcal{H}$ and $\mathcal{S}$ represent the estimated haze density map and semantic information map, respectively. $\mathcal{W}$ denotes the Aggregation Weight Generator, which produces pixel-wise weights with shape $\mathbb{R}^{H \times W}$. The next focus lies in exploring how to effectively estimate $\mathcal{H}$ and $\mathcal{S}$ via a shared backbone.

## 4.2 ESTIMATION OF THE SEMANTIC INFORMATION MAP

Inspired by Figure 1, we leverage CLIP Radford et al. (2021) to jointly extract haze density and semantic information maps. Considering that the pretrained CLIP image encoder is optimized through a classification pretext task, each location in its feature map before pooling captures regional semantic cues Zhang et al. (2022). We therefore directly treat the latent embedding produced by feeding the input image into the encoder as the semantic information map:

$$\mathbf{F}_{\text{img}} = \underbrace{\Phi_{\text{img}}}_{\text{without final pooling}} \big(I_{\text{haze}}\big) \in \mathbb{R}^{H_p \times W_p \times d_e} \tag{3}$$

Here, $\Phi_{\text{img}}$ denotes the CLIP image encoder without its final pooling layer, enabling the generation of a patch-wise embedding. $I_{\text{haze}}$ is the input image to the dehazing network. $H_p$ and $W_p$ denote the height and width of the encoded embedding, and $d_e$ is the hidden dimension.

Using only $\mathbf{F}_{\text{img}}$ yields suboptimal performance because CLIP provides mainly high-level representations that lack low-level semantic details. To address this limitation, we incorporate the input features of the current block, $\mathbf{F}_{\text{in}}$, to complement $\mathbf{F}_{\text{img}}$, and introduce a bidirectional cross-attention mechanism to align their semantic information across scales, producing a refined semantic map $\mathcal{S}$:

$$\mathcal{S} = W\Big[\underbrace{\text{Attn}(Q_{img}, K_{in}, V_{in})}_{\text{high-level query on low-level}} \;\|\; \underbrace{\text{Attn}(Q_{in}, K_{img}, V_{img})}_{\text{low-level query on high-level}}\Big] \tag{4}$$

Here, $Q_i$, $K_i$ and $V_i$ are the query, key, and value derived from $\mathbf{F}_i$ (with $i \in \{img, in\}$) after channel reduction or adaptive pooling, and $\text{Attn}(\cdot)$ denotes the attention operation. $W$ is a projection matrix.

## 4.3 ESTIMATION OF THE HAZE DENSITY MAP

### 4.3.1 WORKFLOW OF THE PROPOSED METHOD

As shown in Figure 1, the similarity between an image embedding and a haze-describing prompt grows monotonically with haze density. Building on this observation, we design a streamlined

estimation pipeline. The input image is first mapped to a latent representation by the pretrained CLIP image encoder (Eq. 3). We then construct a prompt set $T = [T_{\text{haze}}, T_{\text{clear}}]$ (*e.g.*, [*"hazy image"*, *"clear image"*]) and project it into the same space through the CLIP text encoder. Finally, we interpret the similarity between the image embedding and the haze-oriented text embedding as the predicted haze density map.

$$\mathcal{H} = \text{Softmax}\Big(\text{sim}\big(\mathbf{F}_{\text{img}}, \underbrace{\Phi_{\text{txt}}(T)}_{\mathbf{F}_{\text{txt}} \in \mathbb{R}^{2 \times d_c}}\big)\Big)[:,:,0] \in \mathbb{R}^{H_p \times W_p} \tag{5}$$

Here, $\Phi_{\text{text}}$ denotes the CLIP text encoder, and $\text{sim}(\cdot, \cdot)$ is the similarity function.

### 4.3.2 LEARNABLE PROMPT OPTIMIZATION

To improve estimation accuracy and alleviate the burden of laborious prompt engineering, we employ learnable prompt tokens rather than manually defined prompts to represent abstract haze and clear conditions.

Our training procedure comprises two stages. **Stage 1** optimizes learnable paired prompts via a cross-entropy objective, allowing them to preliminarily distinguish between hazy and clear images. During training, a hazy image and a clear image, $I_{\text{haze}}, I_{\text{clear}} \in \mathbb{R}^{H \times W \times 3}$, are used, and the first-stage loss $\mathcal{L}_1$ is defined as:

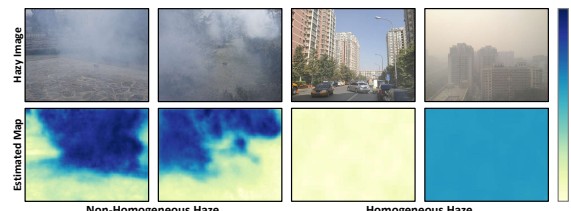

Figure 4: **Illustration of estimated haze density maps**. The proposed method can identify the relative haze density not only in different regions of non-homogeneous haze images but also between two homogeneous haze images.

$$\mathcal{L}_1 = -\Big[y \log \hat{y} + (1-y) \log\big(1 - \hat{y}\big)\Big],$$

$$\text{where} \quad \hat{y} = \frac{\exp\Big(\cos\big(\Phi_{\text{img}}(I), \Phi_{\text{txt}}(T_{\text{clear}})\big)\Big)}{\sum\limits_{i \in \{\text{haze, clear}\}} \exp\Big(\cos\big(\Phi_{\text{img}}(I), \Phi_{\text{txt}}(T_i)\big)\Big)}. \tag{6}$$

where $I \in \{I_{\text{haze}}, I_{\text{clear}}\}$, and $y$ is the corresponding label, with 0 indicating a hazy image and 1 indicating a clear image.

**Stage 2** aims to predict haze density more accurately. The most straightforward and effective optimization approach is regression, yet existing datasets lack ground-truth density maps. To address this, we construct triplets $\{I_{\text{haze}}, I_{\text{clear}}, I_{\text{density}}\}$ using the Atmospheric Scattering Model Narasimhan & Nayar (2002). The second-stage loss is formulated as:

$$\mathcal{L}_2 = \begin{cases} \alpha_1 \text{MSE}\big(\hat{\mathcal{H}}, I_{\text{density}}\big) + \alpha_2 \mathcal{L}_1, & y = 0, \\ \mathcal{L}_1, & y = 1, \end{cases} \tag{7}$$

where $\text{MSE}(\cdot)$ denotes the mean squared error, and $\hat{\mathcal{H}}$ is obtained from $I_{\text{haze}}$ and the learnable paired prompts $T = [T_{\text{haze}}, T_{\text{clear}}]$ according to Equation 5. $\alpha_1$ and $\alpha_2$ are the weights of different loss functions. Training remains lightweight because the CLIP image encoder is not fine-tuned.

Finally, we apply the learned paired prompts to generate the estimated patch-wise haze density map $\mathcal{H}$. As shown in Figure 4, our method provides an effective and general solution applicable to both homogeneous and non-homogeneous haze. Please refer Appendix A for more training details.

### 4.4 IDEALIZED DUAL-STREAM DEHAZING FRAMEWORK WITH AGGREGATOR INTEGRATION

Finally, based on the proposed CLIP-enhanced Dual-path Aggregator, we develop a baseline model, **DehazaMatic**, for the dual-stream dehazing network to further investigate the potential of this class of approaches and to assess their ability to achieve state-of-the-art performance. The detailed architecture of DehazaMatic is presented in Appendix B.

Table 1: **Quantitative results after replacing the naïve aggregator with the proposed CedA.** The extra runtime introduced by CedA was measured on an NVIDIA A100 GPU with 256×256 inputs.

| Methods | SOTS-Outdoor | | SOTS-Indoor | | NH-Haze | | Dense-Haze | | RTTS | | Extra Runtime (ms) |
|---|---|---|---|---|---|---|---|---|---|---|---|
| | PSNR↑ | SSIM↑ | PSNR↑ | SSIM↑ | PSNR↑ | SSIM↑ | PSNR↑ | SSIM↑ | FADE↓ | BRISQUE↓ | |
| FSDGN | 36.95 | 0.986 | 40.12 | 0.990 | 19.51 | 0.712 | 16.42 | 0.556 | 1.492 | 36.218 | 5.1 |
| → CedA | +0.67 | +0.002 | +1.03 | +0.001 | +0.51 | +0.019 | +0.56 | +0.038 | -0.035 | -0.327 | |
| HyLoG-ViT | 36.28 | 0.990 | 39.95 | 0.992 | 21.02 | 0.775 | 16.68 | 0.608 | 1.685 | 37.539 | 4.6 |
| → CedA | +0.81 | +0.002 | +0.50 | +0.002 | +0.11 | +0.003 | +0.34 | +0.012 | -0.047 | -0.237 | |
| Dual-Former | 36.33 | 0.988 | 40.04 | 0.991 | 19.68 | 0.682 | 16.09 | 0.512 | 1.357 | 34.726 | 3.7 |
| → CedA | +1.12 | +0.003 | +0.88 | +0.002 | +0.49 | +0.020 | +0.62 | +0.025 | +0.003 | -0.684 | |

## 5 EXPERIMENTS

### 5.1 DATASETS

We evaluate our method on both synthetic and real-world benchmarks. For synthetic experiments, we consider homogeneous and non-homogeneous haze conditions. For homogeneous haze, we adopt the RESIDE dataset Li et al. (2018), which provides two training partitions: the Indoor Training Set (ITS) with 13,990 paired indoor samples, and the Outdoor Training Set (OTS) with 313,950 paired outdoor samples. Evaluation is carried out on the corresponding splits of the Synthetic Objective Testing Set (SOTS). For non-homogeneous haze, we employ NH-HAZE Ancuti et al. (2020) and Dense-Haze Ancuti et al. (2019), both produced using a professional haze generator to mimic complex real-world scattering. Each dataset contains 55 image pairs, where the final 5 pairs are reserved for testing and the remaining 50 for training. To assess generalization in real scenarios, we use the RTTS dataset Li et al. (2018), comprising 4,322 unpaired hazy images captured in the wild.

### 5.2 EMPIRICAL EVALUATION OF THE CLIP-ENHANCED DUAL-PATH AGGREGATOR

To evaluate its broader applicability, we replace the naïve aggregator in representative dual-stream dehazing networks with the proposed CLIP-enhanced Dual-path Aggregator (CedA) and investigate whether this substitution improves the overall performance of this family of models. We experiment with three methods, each capturing both long- and short-range dependencies: FSDGN Yu et al. (2022) uses frequency-domain modeling and convolution; HyLoG-ViT Zhao et al. (2021) combines pooled self-attention with window-based self-attention; and Dual-Former Chen et al. (2024a) integrates channel attention with window-based self-attention. For fairness, we retrain all models following their original configurations and keep the training settings identical before and after replacing the aggregator with CedA.

As shown in Table 1, substituting the current dual-path aggregator with CedA yields substantial performance gains across multiple datasets, while the additional inference cost introduced by CedA is negligible. This is because, although the integrated CLIP Radford et al. (2021) model contains a large number of parameters, the inference time for a single image is only about 3 ms.

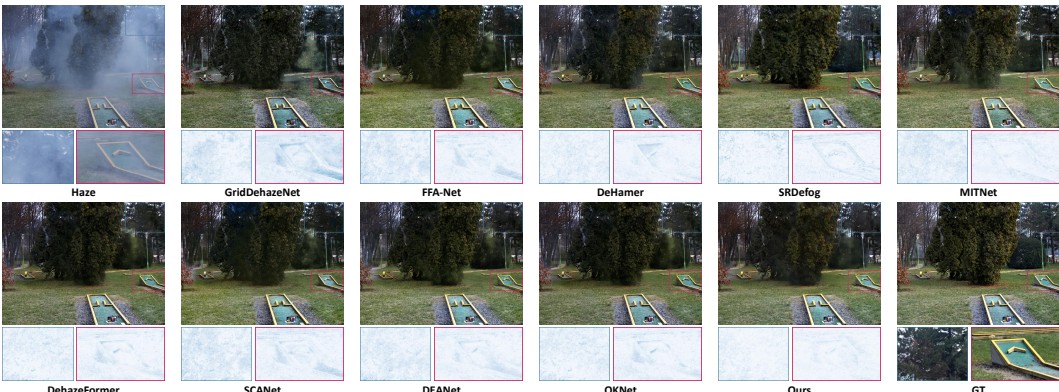

Figure 5: **Visual comparisons on non-homogeneous haze**. The bottom shows enlarged error maps of selected regions, where darker blue indicates larger restoration errors. Please zoom in to view.

Table 2: **Quantitative results on synthetic dehazing benchmarks.** Best results are shown in **bold**.

| Methods | Homogeneous Haze | | | | | | Non-homogeneous Haze | | | | | | Overhead | |
|---|---|---|---|---|---|---|---|---|---|---|---|---|---|---|
| | SOTS-Outdoor | | SOTS-Indoor | | Avg | | NH-Haze | | Dense-Haze | | Avg | | Params | MACs |
| | PSNR↑ | SSIM↑ | PSNR↑ | SSIM↑ | PSNR↑ | SSIM↑ | PSNR↑ | SSIM↑ | PSNR↑ | SSIM↑ | PSNR↑ | SSIM↑ | | |
| DCP He et al. (2010) | 19.14 | 0.861 | 16.61 | 0.855 | 17.88 | 0.858 | 10.57 | 0.520 | 10.06 | 0.385 | 10.32 | 0.453 | - | - |
| AOD-Net Li et al. (2017) | 24.14 | 0.920 | 20.51 | 0.816 | 22.33 | 0.868 | 15.40 | 0.569 | 13.14 | 0.414 | 14.27 | 0.492 | 1.76K | 0.12G |
| GridDehazeNet Liu et al. (2019) | 30.86 | 0.982 | 32.16 | 0.984 | 31.51 | 0.983 | 18.33 | 0.667 | 14.96 | 0.533 | 16.65 | 0.600 | 0.96M | 21.55G |
| FFA-Net Qin et al. (2020) | 33.57 | 0.984 | 36.39 | 0.989 | 34.98 | 0.987 | 19.87 | 0.692 | 16.09 | 0.503 | 17.98 | 0.598 | 4.46M | 288.86G |
| DeHamer Guo et al. (2022) | 35.18 | 0.986 | 36.63 | 0.988 | 35.91 | 0.987 | 20.66 | 0.684 | 16.62 | 0.560 | 18.64 | 0.622 | 132.45M | 59.25G |
| SRDefog Jin et al. (2022) | - | - | - | - | - | - | 20.99 | 0.610 | 16.67 | 0.500 | 18.83 | 0.555 | 12.56M | 24.18M |
| MAXIM-2S Tu et al. (2022) | 34.19 | 0.985 | 38.11 | 0.991 | 36.15 | 0.988 | - | - | - | - | - | - | 14.10M | 216.00G |
| SGID-PFF Bai et al. (2022) | 30.20 | 0.975 | 38.52 | 0.991 | 34.36 | 0.983 | - | - | - | - | - | - | 13.87M | 156.67G |
| PMNet Ye et al. (2022) | 34.74 | 0.985 | 38.41 | 0.990 | 36.58 | 0.988 | 20.42 | 0.730 | 16.79 | 0.510 | 18.61 | 0.620 | 18.90M | 81.13G |
| MB-TaylorFormer-B Qiu et al. (2023) | 37.42 | 0.989 | 40.71 | 0.992 | 39.07 | 0.991 | - | - | 16.66 | 0.560 | - | - | 2.68M | 38.50G |
| MITNet Shen et al. (2023) | 35.18 | 0.988 | 40.23 | 0.992 | 37.71 | 0.990 | 21.26 | 0.712 | 16.97 | 0.606 | 19.12 | 0.659 | 2.73M | 16.42G |
| DehazeFormer Song et al. (2023) | 34.29 | 0.983 | 38.46 | 0.994 | 36.38 | 0.989 | 20.31 | 0.761 | 16.66 | 0.595 | 18.49 | 0.595 | 4.63M | 48.64G |
| SCANet Guo et al. (2023) | - | - | - | - | - | - | 19.52 | 0.649 | 15.35 | 0.508 | 17.44 | 0.579 | 2.39M | 258.63G |
| DEANet Chen et al. (2024b) | 36.03 | 0.989 | 40.20 | 0.993 | 38.12 | 0.991 | 20.84 | 0.801 | 16.73 | 0.602 | 18.79 | 0.702 | 3.65M | 32.23G |
| UVM-Net Zheng & Wu (2024) | 34.92 | 0.984 | 40.17 | **0.996** | 37.55 | 0.990 | - | - | - | - | - | - | 19.25M | 173.55G |
| OKNet Cui et al. (2024) | 35.45 | 0.992 | 37.59 | 0.994 | 36.52 | 0.993 | 20.29 | 0.800 | 16.85 | 0.620 | 18.57 | 0.710 | 4.42M | 39.54G |
| DCMPNet Zhang et al. (2024b) | 36.56 | 0.993 | **42.18** | **0.996** | 39.37 | 0.995 | - | - | - | - | - | - | 18.59M | 80.42G |
| DehazeMatic | **38.21** | **0.995** | 41.50 | **0.996** | **39.86** | **0.996** | **21.47** | **0.806** | **17.28** | **0.629** | **19.38** | **0.718** | 4.58M | 35.50G |

Table 3: **Quantitative results on real haze**.

| Methods | FADE | BRISQUE | NIMA |
|---|---|---|---|
| PSD | 0.920 | 27.713 | 4.598 |
| D4 | 1.358 | 33.210 | 4.484 |
| DGUN | 1.111 | 27.968 | 4.653 |
| RIDCP | 0.944 | 17.293 | 4.965 |
| CORUN | 0.824 | 11.956 | 5.342 |
| SGDN | 0.873 | 11.549 | 5.128 |
| DehazeMatic | **0.796** | **11.435** | **5.510** |

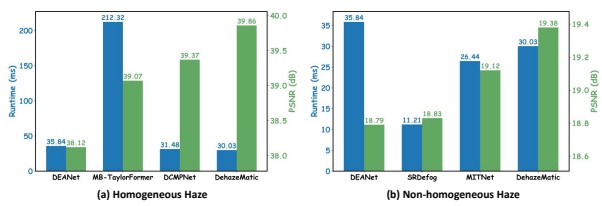

Figure 6: **Performance–runtime trade-off.** Efficiency rises as green bars exceed blue.

## 5.3 EMPIRICAL EVALUATION OF DEHAZEMATIC

We empirically assess whether the proposed CedA module can enable dual-stream dehazing methods to achieve SOTA performance by comparing DehazeMatic with various existing approaches.

**Training Details.** DehazeMatic is implemented with PyTorch on NVIDIA A100 GPUs. We use Adam Kingma & Ba (2014) optimizer with default parameters ($\beta_1 = 0.9$, $\beta_2 = 0.99$) and a cosine annealing strategy Loshchilov & Hutter (2016) with restarts. The initial learning rate is set to $2 \times 10^{-4}$, gradually decreasing to $2 \times 10^{-6}$. We train the homogeneous haze dataset for 200 epochs and the non-homogeneous haze dataset for 400 epochs. The images are randomly cropped to a size of $256 \times 256$ and augmented with flipping. We use L1 loss and perceptual loss Johnson et al. (2016) to supervise dehazing process.

### 5.3.1 PERFORMANCE ON SYNTHETIC HAZE

As shown in Table 2, both DCMPNet Zhang et al. (2024b) and MB-TaylorFormer Qiu et al. (2023) are competitive approaches on synthetic homogeneous haze; however, our DehazeMatic achieves the best overall performance. DCMPNet leverages depth information as guidance, whereas MB-TaylorFormer only employs linear self-attention for feature extraction. These findings highlight the advantage of jointly exploiting haze density and semantic maps for guidance while capturing both short- and long-range dependencies. For synthetic non-homogeneous haze, MITNet Shen et al. (2023) and OKNet Cui et al. (2024) achieve competitive PSNR and SSIM, respectively; nevertheless, DehazeMatic consistently surpasses them, delivering state-of-the-art results across all metrics.

The visual comparison, as shown in Figure 5, further reveals that our approach best preserves fine structures in the blue-boxed tree region and yields the smallest errors along object boundaries in the red-boxed target area.

### 5.3.2 PERFORMANCE ON REAL-WORLD HAZE

To assess the practicality and generalization capability of our model in real-world scenarios, we conduct experiments on the RTTS dataset Li et al. (2018). For fairness, the experimental settings follow those of CORUN Fang et al. (2024a). As shown in Table 3, our approach surpasses all competing methods on no-reference metrics, highlighting its effectiveness.

### 5.3.3 TRADE-OFF BETWEEN PERFORMANCE AND RUNTIME

We further assess efficiency by measuring the runtime on an NVIDIA A100 GPU and comparing the performance–runtime trade-off with the three strongest baselines (Figure 6). Our model runs in only 30.03 ms and achieves 33 frames per second (FPS), meeting real-time processing requirements while offering the best balance between accuracy and speed.

## 5.4 ABLATION STUDIES

We perform ablation studies to validate the contribution of each component. For fairness, we tune the hyperparameters of all variants so that their computational overhead matches that of DehazeMatic.

**(a) Dual-Dependencies.** To assess the effectiveness of the dual-stream design, we build variants that remove either the long-range or short-range path. Results in Table 4 show that combining these complementary dependencies markedly enhances image dehazing.

**(b) Remove CedA.** We replace CedA with simple addition and concatenation to verify the necessity of adaptively integrating short- and long-range dependencies in the dual paths.

**(c) Overall Design.** We further assess the benefit of jointly leveraging haze density and semantic maps for aggregation guidance by removing each component in turn.

**(d) Density Map.** Next, we examine the estimated haze density map. We first evaluate the rationale for using haze density rather than other common

Table 4: Ablation experiments of various components of DehazeMatic.

| Setting | | SOTS-Indoor | | NH-Haze | |
|---|---|---|---|---|---|
| | | PSNR↑ | SSIM↑ | PSNR↑ | SSIM↑ |
| (a) Dual-Dependencies | w/o Short-range | 39.78 | 0.994 | 20.56 | 0.779 |
| | w/o Long-range | 38.41 | 0.992 | 20.40 | 0.771 |
| (b) Remove CedA | Addition | 39.55 | 0.993 | 20.71 | 0.781 |
| | Concatenation | 39.80 | 0.994 | 20.74 | 0.784 |
| (c) Overall Design | W/o Density Map | 40.88 | 0.995 | 20.91 | 0.792 |
| | W/o Semantic Map | 41.02 | 0.995 | 21.23 | 0.794 |
| (d) Density Map | Transmission Map | 40.97 | 0.995 | 21.27 | 0.797 |
| | Depth Map | 41.16 | 0.996 | 21.28 | 0.795 |
| | Predefined Prompts | 40.24 | 0.994 | 20.82 | 0.789 |
| (e) Semantic Map | w/o High-level | 41.12 | 0.995 | 21.30 | 0.800 |
| | w/o Low-level | 41.19 | 0.996 | 21.34 | 0.802 |
| DehazeMatic | | **41.50** | **0.996** | **21.47** | **0.806** |

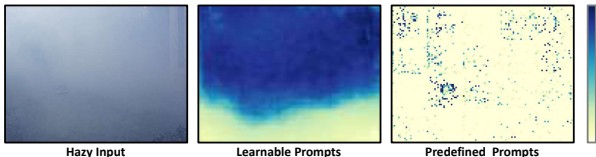

Figure 7: Visual comparison of density maps estimated by learned prompts and manually predefined prompts.

guidance signals in dehazing (*e.g.*, the transmission map from DCP He et al. (2010) or the depth map from Depth Anything Yang et al. (2024a)) for aggregation. Results show that haze density is a more appropriate cue. We then replace learnable prompts with manually predefined ones to assess their effectiveness. As shown in Table 4, predefined prompts cause a marked performance decline, even falling below the variant that relies solely on semantic maps. Moreover, Figure 7 shows that predefined prompts fail to produce valid patch-wise haze density maps.

**(e) Semantic Map.** Finally, we validate the necessity of each type of information in the refined semantic map by ablating either high-level semantic information (*i.e.*, from CLIP) or low-level semantic information (*i.e.*, input of Tramba blocks).

# 6 CONCLUSION

In this paper, we first highlight the limitations of existing dual-stream dehazing methods, namely the lack of clear guidance on how to balance the relative importance of short- and long-range dependencies. Through extensive quantitative and qualitative analyses, we identify haze density and semantic information as critical factors. Then, we propose the CLIP-enhanced Dual-path Aggregator (CedA), a plug-and-play module designed to replace naïve aggregators in existing networks. CedA leverages a shared backbone to efficiently estimate haze density and semantic maps, subsequently generating reliable aggregation weights. This framework also enables, for the first time, accurate estimation of haze density maps. Finally, building on CedA, we present DehazeMatic, a benchmark dual-stream dehazing network, and demonstrate that it achieves SOTA performance in multiple datasets, underscoring the untapped potential of dual-stream architectures for haze removal.

## ETHICS STATEMENT

This work complies with the ICLR Code of Ethics. The datasets used in our experiments are publicly available and do not contain any personally identifiable or sensitive information. Our research does not involve human subjects, animal studies, or any sensitive social data. We believe our findings do not pose direct ethical risks.

## REPRODUCIBILITY STATEMENT

We have taken several measures to ensure reproducibility. The architecture details and evaluation protocols are provided in Section 4, Section 5, and Appendix B. Additional implementation details, including data preprocessing steps and training hyperparameters, are included in Section 5 and Appendix A. A thorough presentation of the experimental results and analyses can be found in Section 5. Although we do not release code at this stage, these details should allow independent researchers to reproduce our results, and we will make the code publicly available upon acceptance.

## THE USE OF LARGE LANGUAGE MODELS

In preparing this manuscript, we employed a large language model (ChatGPT, OpenAI) solely as a general-purpose assistant to improve the clarity and grammar of the text. The model was not involved in research ideation, experimental design, data analysis, or interpretation of results. All scientific content, methodologies, and conclusions were developed entirely by the authors. The authors take full responsibility for the final content of the paper.

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

APPENDIX

# A    LEARNABLE HAZE/CLEAR PROMPTS

## A.1    DEFINITION OF HAZE DENSITY MAP

The Atmospheric Scattering Model (ASM) Narasimhan & Nayar (2002) is defined as:

$$\mathbf{I}(\mathbf{x}) = \mathbf{J}(\mathbf{x})t(\mathbf{x}) + \mathbf{A}(1 - t(\mathbf{x})),$$
$$t(\mathbf{x}) = e^{-\beta d(\mathbf{x})}. \tag{8}$$

Here, $\mathbf{x} = (x, y)$ is a 2D vector representing the pixel coordinates in the image. $\mathbf{I}$ denotes the observed hazy image, while $\mathbf{J}$ represents the scene radiance image, typically regarded as the clear image. $\mathbf{A}$ is the global atmospheric light, often considered to approximate the color of the sky, atmosphere, or horizon. $t$ is the transmission map, which is a scalar within the range $[0, 1]$.

According to Equation 8, transmission map $t$ depends on the atmospheric scattering coefficient $\beta$ and the scene depth $d$. $\beta$ is typically defined as a global constant to characterize homogeneous haze scene. However, in reality, particularly in outdoor environments, most haze is non-homogeneous (*e.g.*, haze on highways), the scattering coefficient $\beta$ should also be treated as non-homogeneous. Therefore, $t(\mathbf{x})$ in Equation 8 can be rewritten as:

$$t(\mathbf{x}) = e^{-\beta(\mathbf{x})d(\mathbf{x})}. \tag{9}$$

The scattering coefficient $\beta$ is determined by the physical properties of the atmosphere (*e.g.*, particulate matter, size, shape, and concentration) and most directly reflects the haze density, so we treat spatial variables $\beta$ as the haze density map.

## A.2    TRAINING PROCESS

### A.2.1    GENERATION OF TRAINING DATA

To directly constrain the estimated haze density map in a regression manner and thereby train the learnable haze/clear prompts, we first need triplet data $\{I_{\text{haze}}, I_{\text{clear}}, I_{\text{density}}\}$ that includes the ground truth haze density map $I_{\text{density}}$. We implement this based on the ASM.

We use the clear images from the training set of the RESIDE dataset Li et al. (2018) as $I_{\text{clear}}$ and given $I_{\text{density}}$, then generate $I_{\text{haze}}$ by utilizing these two to construct triplet data. The depth map $d$ required by ASM is obtained by inputting $I_{\text{clear}}$ into the Depth Anything Yang et al. (2024a) model. To ensure the learned prompts are applicable to both homogeneous and non-homogeneous haze, the generated dataset should include both types of haze, equivalent to providing homogeneous and non-homogeneous density maps.

Providing homogeneous density maps is easy. We simply assign a global constant to $\beta$. However, it is difficult to obtain the non-homogeneous density maps required to synthesize non-homogeneous hazy images. We propose obtaining these density maps from remote sensing (RS) non-homogeneous hazy images, as the scene depth of each pixel in RS images can be approximately considered consistent. In this case, the transmission map derived from a prior (such as the dark channel prior) can be treated as an approximate density map. However, this method still introduces minor interference from scene textures. To mitigate this, we apply smoothing techniques for post-processing. Using this approach, we generate 20,000 non-homogeneous haze density maps and randomly sample from them when synthesizing hazy images. For each clear image, we generate three images with different haze density.

### A.2.2    IMPLEMENTATION DETAILS

For both training stages we use the Adam Kingma & Ba (2014) optimizer with its default parameters ($\beta_1 = 0.9$, $\beta_2 = 0.99$) and the cosine annealing strategy Loshchilov & Hutter (2016). The initial learning rate of the first stage is set $2 \times 10^{-5}$, gradually decreasing to $2 \times 10^{-6}$. The batch size is 4, and it only lasts for 1 epoch.

In the second stage of training, the input data is divided into two types: $[I_{\text{clear}}]$ and $[I_{\text{haze}}, I_{\text{density}}]$, which appear randomly within each batch. The initial learning rate of the first stage is set $1 \times 10^{-5}$, gradually decreasing to $1 \times 10^{-6}$. The batch size is 4, and training lasts for 30 epochs.

## B ARCHITECTURE OF DEHAZAMATIC

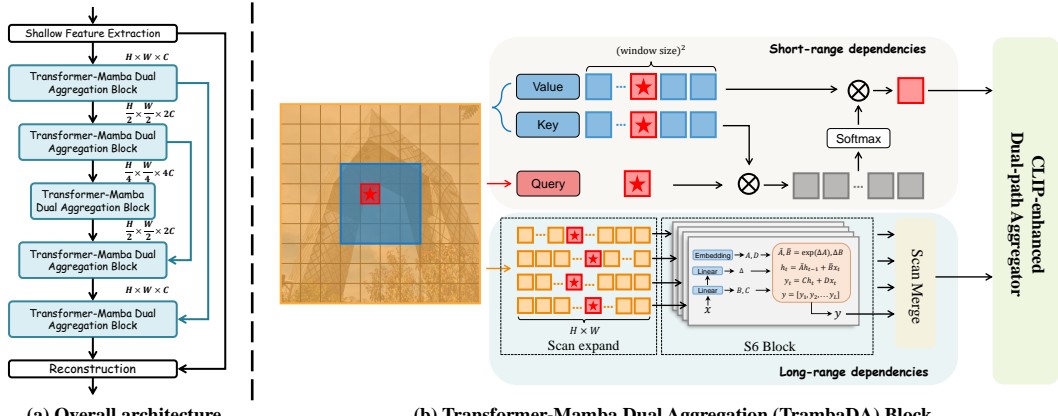

**(a) Overall architecture**       **(b) Transformer-Mamba Dual Aggregation (TrambaDA) Block**

Figure 8: (a) Overall architecture of DehazeMatic. (b) Internal design of the Transformer–Mamba Dual Aggregation (TrambaDA) block.

The overall architecture of DehazeMatic is illustrated in Figure 8. Starting from a hazy input, shallow features are extracted and subsequently processed by a symmetric encoder–decoder framework. Each encoder and decoder stage is composed of several Transformer–Mamba Dual Aggregation (TrambaDA) blocks together with appropriate downsampling or upsampling layers. Skip connections are introduced at every resolution level to facilitate gradient propagation and feature reuse. The output from the final decoding stage is fused with the original hazy image through a residual pathway, producing the haze-free image.

### B.1 CAPTURING SHORT-RANGE DEPENDENCIES

We construct this path using window-based self-attention Liu et al. (2021), which offers stronger fitting capability through dynamic weights. Given an input feature map $F_{\text{in}} \in \mathbb{R}^{H \times W \times C}$, we partition it into $N = HW/M^2$ non-overlapping windows of size $M \times M$. For window $i$, the flattened feature is denoted as $F_{\text{in}}^{(i)} \in \mathbb{R}^{M^2 \times C}$. Assuming a single attention head, self-attention is computed as:

$$Q^{(i)} = F_{\text{in}}^{(i)} W_Q, \quad K^{(i)} = F_{\text{in}}^{(i)} W_K, \quad V^{(i)} = F_{\text{in}}^{(i)} W_V,$$

$$F_{\text{out}}^{(i)} = \text{Softmax}\left(\frac{Q^{(i)} K^{(i)\top}}{\sqrt{d_k}}\right) V^{(i)}, \tag{10}$$

$$F_{\text{out}}^{\text{short}} = \left\{ F_{\text{out}}^{(1)}, F_{\text{out}}^{(2)}, \ldots, F_{\text{out}}^{(N)} \right\},$$

where $W_Q, W_K, W_V \in \mathbb{R}^{C \times d_k}$ are learnable projection matrices.

### B.2 CAPTURING LONG-RANGE DEPENDENCIES

To capture global interactions with linear complexity, we incorporate Mamba's S6 block Gu & Dao (2023). Because visual data are non-causal, directly applying S6 on a flattened feature map can introduce directional bias Liu et al. (2024b). Following Vmamba Liu et al. (2024b), we unfold the feature map along four scanning directions to form sequences $\{F_{\text{in}}^d\}_{d=1}^4$. Each sequence is processed

by an S6 operator, and the results are merged:

$$\begin{aligned}
F_{\text{in}}^d &= \text{Expand}(F_{\text{in}}, d), \quad d \in \{1, 2, 3, 4\}, \\
\bar{F}^d &= \text{S6}\big(F_{\text{in}}^d\big), \\
F_{\text{out}}^{\text{long}} &= \text{Merge}\big(\bar{F}^1, \bar{F}^2, \bar{F}^3, \bar{F}^4\big).
\end{aligned} \tag{11}$$

Here, $\text{Expand}(\cdot)$ and $\text{Merge}(\cdot)$ denote the scan-expand and scan-merge procedures.

## C  MORE VISUAL COMPARISONS

### C.1  VISUAL COMPARISONS ON RESIDE

Visual comparisons on the RESIDE Li et al. (2018) SOTS Indoor and Outdoor datasets are shown in Figures 9 and 10.

### C.2  VISUAL COMPARISONS ON NH-HAZE

Figure 11 shows the visual comparisons on the NH-Haze Ancuti et al. (2020) dataset.

### C.3  VISUAL COMPARISONS ON DENSE-HAZE

Visual comparisons on the Dense-Haze Ancuti et al. (2019) dataset are shown in Figure 12. It is evident that our method greatly outperforms all compared methods, achieving the greatest detail restoration, the highest visual quality improvement, and the least haze residual.

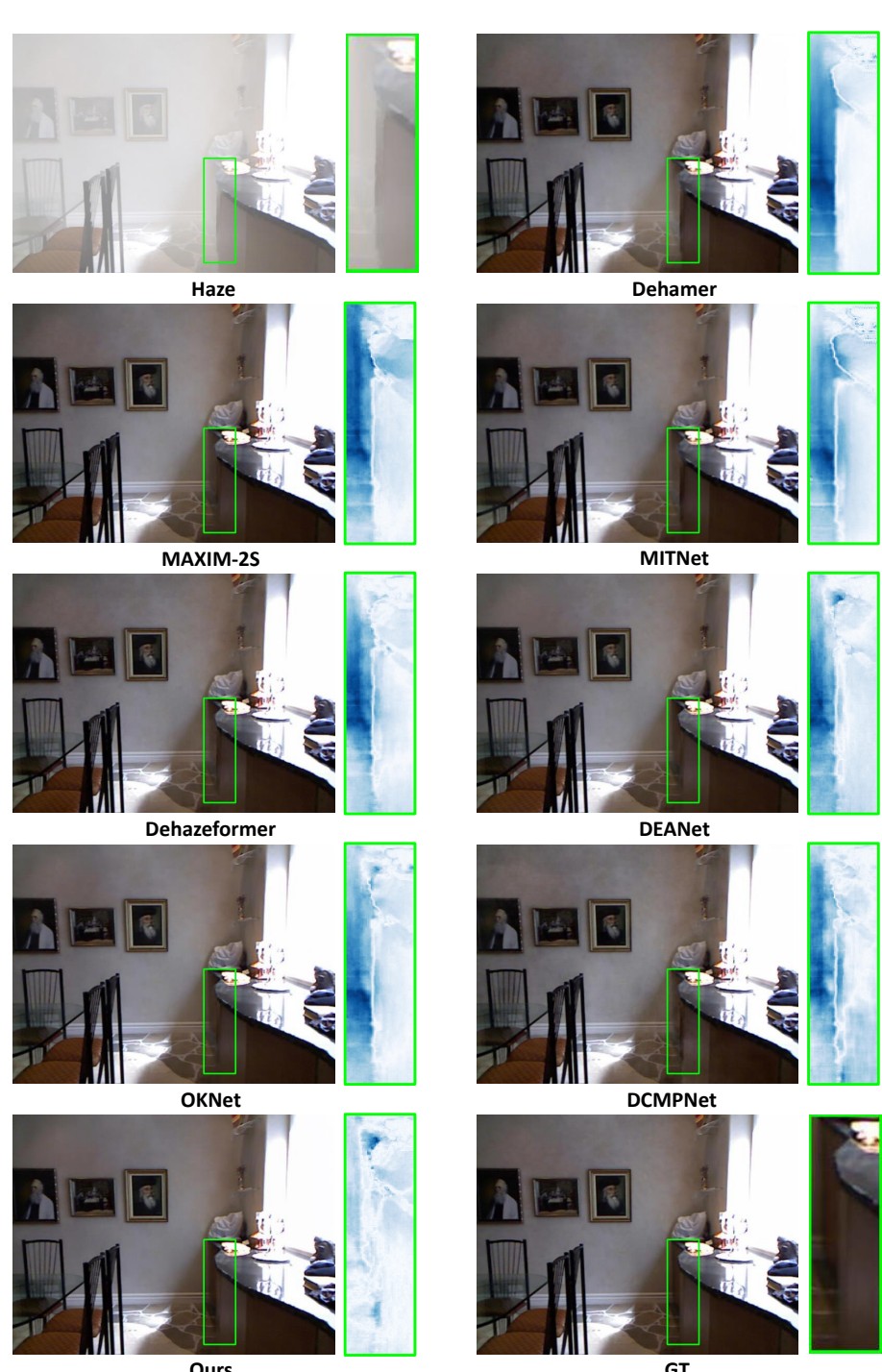

Figure 9: **The qualitative comparison on the RESIDE SOTS-Indoor Li et al. (2018) dataset.**

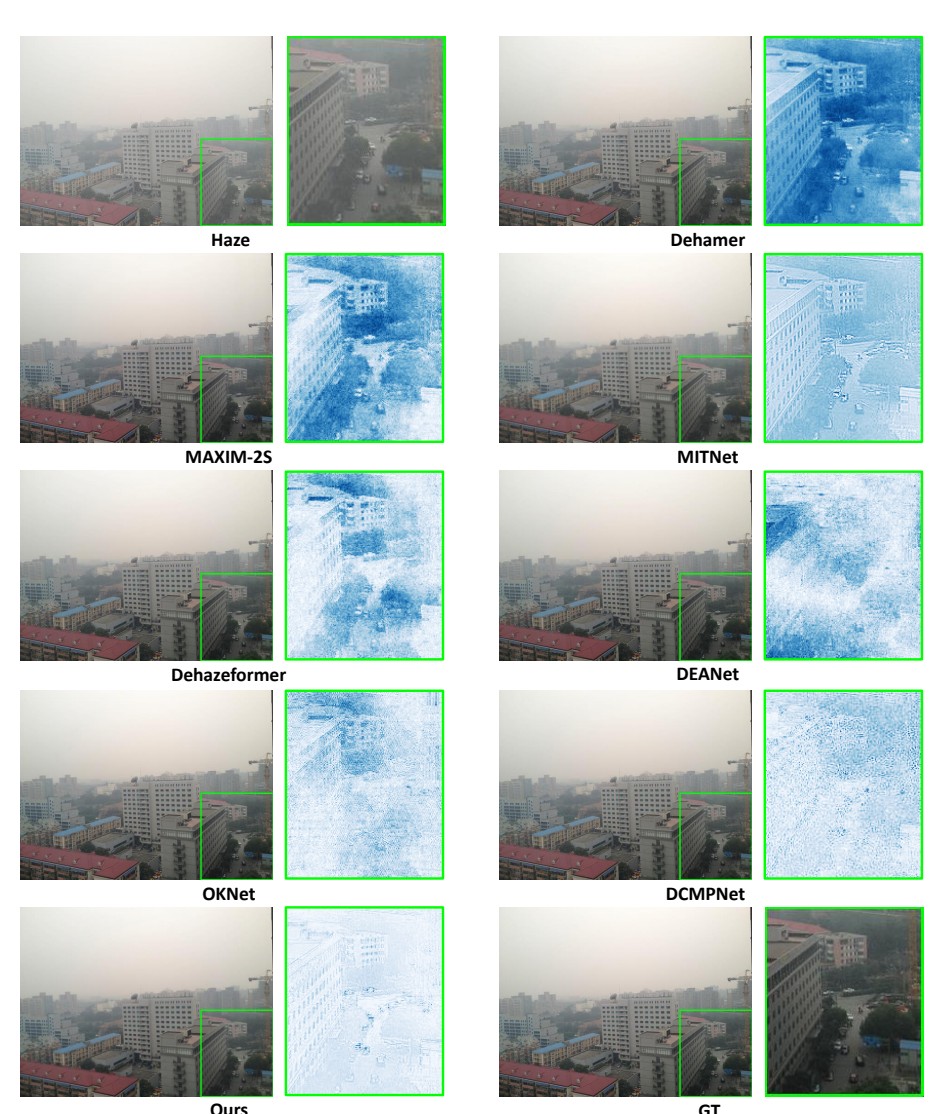

Figure 10: **The qualitative comparison on the RESIDE SOTS-Outdoor Li et al. (2018) dataset.**

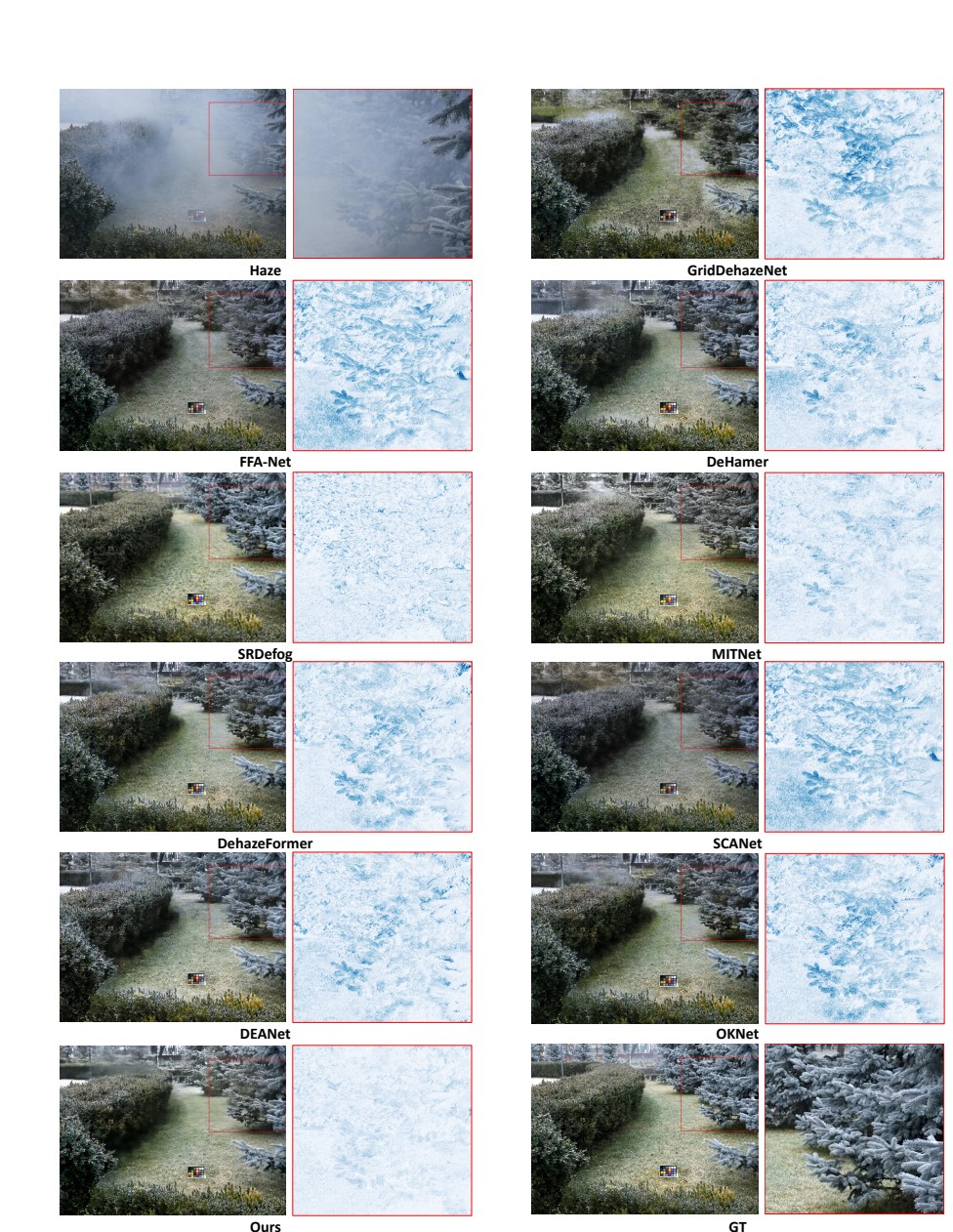

Figure 11: **The qualitative comparison on the NH-HAZE Ancuti et al. (2020) dataset.**

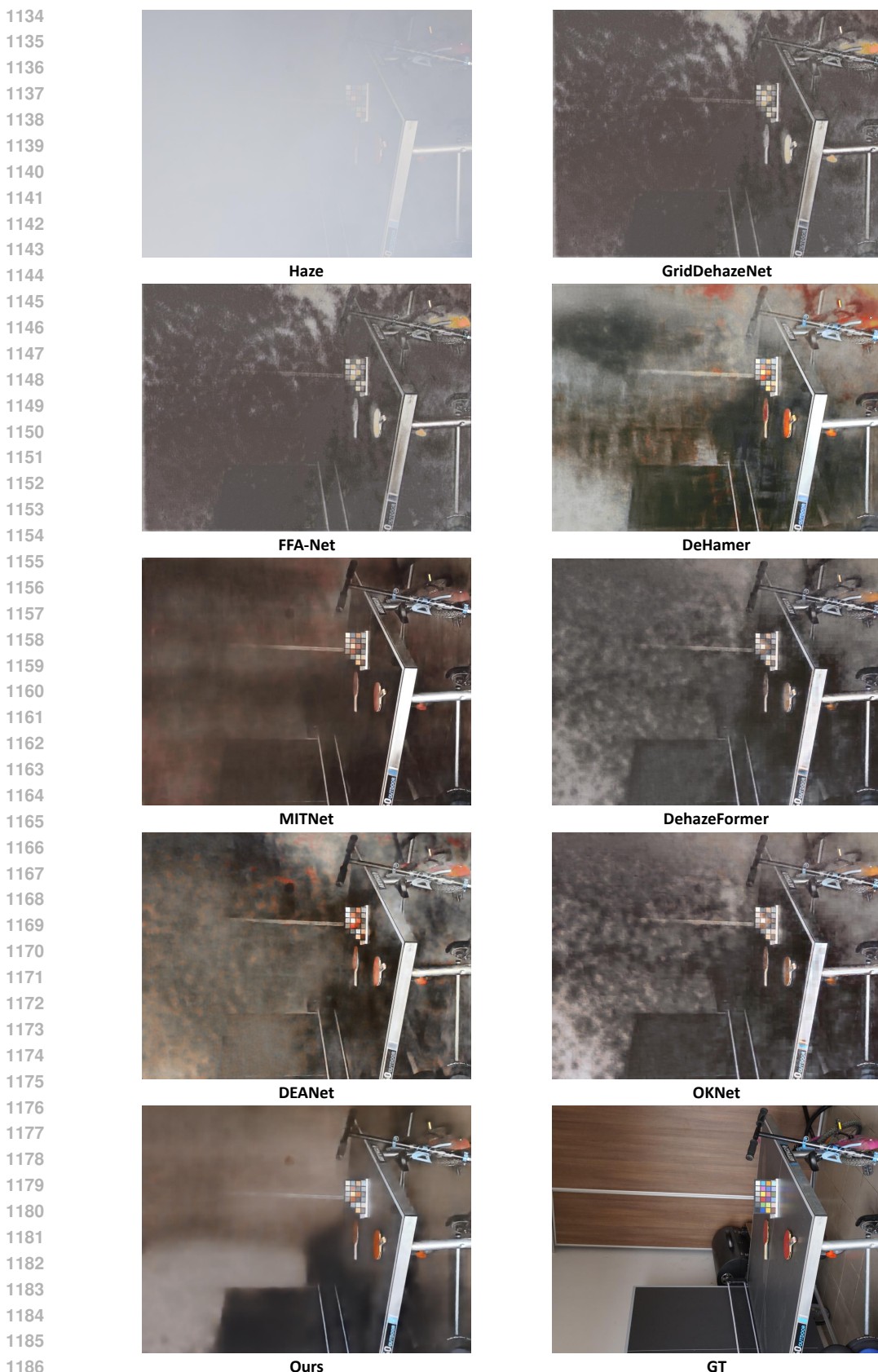

Figure 12: **The qualitative comparison on the Dense-Haze Ancuti et al. (2019) dataset.**

