# OpenReview forum: "Boosting Image Dehazing via Elaborate Integration of Complementary Dependencies"
_ICLR.cc/2026/Conference — ICLR 2026 Conference Withdrawn Submission_

### Official Review · Reviewer_c4TU · 2025-10-25

**Soundness:** 2
**Presentation:** 1
**Contribution:** 2
**Rating:** 2
**Confidence:** 5

**Summary:**

This paper introduces a CLIP-enhanced Dual-Path Aggregator (CedA) that adaptively fuses the two types of dependencies. Through both quantitative and qualitative analyses, the authors identify haze density and semantic information as the two key factors determining the relative importance of these dependencies. CedA leverages a pretrained CLIP model to generate pixel-level haze density and semantic maps, which guide the adaptive fusion process. Based on CedA, the authors further build a new model, DehazeMatic, which achieves state-of-the-art performance across several synthetic and real-world benchmarks with minimal computational overhead.

**Strengths:**

The proposed method generates patch-level haze density maps via CLIP, introducing a meaningful physical prior for dehazing.

It achieves consistent improvements across five major benchmarks, demonstrating the generality and robustness of the proposed module.

**Weaknesses:**

In Figure 4, the authors present an “Illustration of estimated haze density maps,” yet this result appears physically unreasonable. For the Homogeneous Haze case, nearby buildings are visibly clearer, indicating lower haze density, while distant buildings are more obscured, implying higher density. However, the predicted haze density maps display a uniform distribution. Given that haze density estimation is one of the key modules, this discrepancy is concerning.

Table 1 only reports “Extra Runtime” without providing model parameters or GPU memory usage, and Sec. 5.3.3 does not specify image resolution or batch size, making efficiency comparison unclear.

The paper lacks interpretability analysis of the CedA aggregation weight generator. In Eq. (2) and Fig. 3, the Aggregation Weight Generator W(H,S) is defined merely as a lightweight linear layer, without any visualization or sensitivity analysis.

Since many more advanced vision-language models (VLMs) and CLIP variants have emerged, the paper should clarify why CLIP was specifically chosen instead of newer alternatives.

In Sec. 4.3.2, the authors train on synthetic “pseudo-density maps” generated by the Atmospheric Scattering Model (ASM) but never validate their correlation with real haze density. Because ASM-generated haze can differ significantly from real-world haze, the learned model may overfit statistical artifacts rather than true physical characteristics.

Eq. (1) lacks an explanation of the statistical assumptions and variance analysis for the expectation term; in Eq. (5), the meaning of the Softmax output channel “[:, :, 0]” is not defined; and Eqs. (6–7) introduce a joint optimization objective without any convergence or stability analysis.

**Questions:**

See Weakness

---

### Official Review · Reviewer_oBiT · 2025-10-26

**Soundness:** 2
**Presentation:** 2
**Contribution:** 2
**Rating:** 4
**Confidence:** 4

**Summary:**

This paper presents a novel image dehazing framework that integrates short-range and long-range dependencies through a CLIP-enhanced Dual-path Aggregator. The method utilizes CLIP priors to generate semantic and haze density maps, which are used to adaptively guide feature aggregation in dual-stream architectures. The authors further build a benchmark model, DehazeMatic. The main contributions include identifying the key factors influencing dependency integration, proposing a plug-and-play aggregation module, and achieving accurate haze density estimation for the first time.

**Strengths:**

1. The paper identifies haze density and semantic information as key factors governing the balance between short-range and long-range dependencies and develops a CLIP-guided aggregation strategy, going beyond naive addition or concatenation used in prior work.
2. This provides a new interpretable dimension for dehazing research and may inspire further method development grounded in physical properties.
3. The paper includes extensive evaluations on synthetic and real-world datasets, ablation studies, visual comparisons, and runtime analysis, supporting both effectiveness and efficiency.

**Weaknesses:**

1. Although the paper provides empirical observations, it lacks a deeper theoretical justification for why CLIP embeddings effectively guide dependency integration.
2. The evaluation on real-world datasets does not include comparisons with diffusion-based models or large-scale foundation models, which are becoming increasingly relevant in this domain.
3. While the authors state that CLIP introduces negligible inference cost, they do not report training time, memory usage, or FLOPs, limiting the credibility of the claimed efficiency.
4. The contributions rely on integrating existing dual-stream architectures with CLIP-guidance rather than introducing a fundamentally new dehazing mechanism.

**Questions:**

See the above parts.

---

### Official Review · Reviewer_7n8T · 2025-10-28

**Soundness:** 3
**Presentation:** 3
**Contribution:** 2
**Rating:** 4
**Confidence:** 4

**Summary:**

This paper investigates how to effectively integrate short-range and long-range dependencies in dual-stream image dehazing networks. Through detailed empirical analysis, the authors reveal that both haze density and semantic information jointly determine the importance allocation between these dependencies. Based on this insight, the authors propose a CLIP-enhanced Dual-path Aggregator (CedA), which leverages a frozen CLIP image encoder with learnable prompts to generate complementary semantic and fine-grained haze density maps. These maps are then projected to produce pixel-wise weights that adaptively fuse the outputs of the short- and long-range streams. Extensive experiments on both synthetic and real-world datasets demonstrate that integrating CedA consistently boosts performance across various dual-stream frameworks, achieving state-of-the-art PSNR and SSIM results.

**Strengths:**

1. This paper is the first quantitatively demonstrate that haze density and semantic information are the key factors influencing the relative importance between short-range and long-range dependencies in image dehazing.
2. By leveraging prompt tuning, the authors effectively adapt CLIP to a low-level vision task and design a plug-and-play module that can be seamlessly integrated into any existing dual-stream framework.
3. Experimental results show that the proposed method achieves superior dehazing performance, consistently outperforming strong baselines across multiple synthetic and real-world datasets.

**Weaknesses:**

1. The method relies solely on prompt tuning for adapting CLIP, which inherently limits its performance to the representational capacity of the pretrained CLIP model. The authors should further evaluate the approach on out-of-distribution or extreme scenarios that CLIP was not exposed to during pretraining.
2. The model is trained exclusively on synthetic hazy images, raising concerns about its generalization to real-world scenes. The authors should clarify whether the synthetic-to-real gap affects performance and why real-world data were not utilized for training.
3. The motivation and theoretical explanation behind the proposed design are not sufficiently elaborated. A deeper discussion of the underlying principles and empirical observations would make the rationale of the method more convincing.

**Questions:**

Please refer to my comments on weaknesses.

---

### Official Review · Reviewer_ND7M · 2025-11-01

**Soundness:** 3
**Presentation:** 3
**Contribution:** 3
**Rating:** 6
**Confidence:** 4

**Summary:**

- The paper’s core idea is to obtain haze density (H) via CLIP and semantics (S) via cross-attention, then perform per-token fusion (W(H,S)) to adaptively combine Short/Long streams at the pixel level.
- Experiments replace naive aggregators in existing dual-stream backbones with CedA (plug-in) and show consistent gains; the integrated model (DehazeMatic) also reports strong overall results.
- Prompts are trained in two stages (CE classification → synthetic regression) to predict patch-wise density, and S is formed by bidirectional cross-attention between CLIP embeddings and low-level features.
- The framework emphasizes practicality, but the overall training objective (total loss) and gradient flow are not explicitly formalized in the main text.
- The synthetic data pipeline is strong, so it is unclear how much of the reported gains come from CedA itself versus the pipeline.

**Strengths:**

- Per-token fusion justified by H·S mitigates dual-stream limitations and improves interpretability.
- Plug-and-play validation demonstrates backbone-independent gains.
- Learnable prompts surpass zero-shot for density map quality; visualizations are intuitive.
- Integration into DehazeMatic supports the work’s practical value.
- Ablation shows performance drops when removing H/S, partially evidencing the contribution of the key components.
- Although haze density is trained on synthetic data, the method shows consistent improvements on real datasets.

**Weaknesses:**

- The total loss and gradient flow are not clearly summarized in the main paper.
- Insufficient separation of data-pipeline effects.
They generate 20k non-homogeneous density maps from remote-sensing images using DCP plus smoothing for synthesis (Appendix A.2.1). This strong synthetic pipeline could substantially determine learnable prompt (H) quality. However, the main text/experiments do not assess the pipeline's sensitivity or necessity—e.g., performance without it or with alternatives.

**Questions:**

Could you specify in the main text the end-to-end total loss, the H/S→W spatial alignment, and the gate activation (sigmoid/softmax, temperature)?

---

### Note · Authors · 2025-12-08

I have read and agree with the venue's withdrawal policy on behalf of myself and my co-authors.